# Lynch Syndrome: Its Impact on Urothelial Carcinoma

**DOI:** 10.3390/ijms22020531

**Published:** 2021-01-07

**Authors:** Andrea Katharina Lindner, Gert Schachtner, Gennadi Tulchiner, Martin Thurnher, Gerold Untergasser, Peter Obrist, Iris Pipp, Fabian Steinkohl, Wolfgang Horninger, Zoran Culig, Renate Pichler

**Affiliations:** 1Department of Urology, Medical University Innsbruck, 6020 Innsbruck, Austria; andrea.lindner@i-med.ac.at (A.K.L.); Gert.Schachtner@tirol-kliniken.at (G.S.); Gennadi.Tulchiner@i-med.ac.at (G.T.); martin.thurnher@i-med.ac.at (M.T.); Wolfgang.Horninger@i-med.ac.at (W.H.); Zoran.Culig@i-med.ac.at (Z.C.); 2Immunotherapy Unit, Department of Urology, Medical University Innsbruck, 6020 Innsbruck, Austria; 3Department of Internal Medicine V, Medical University Innsbruck, 6020 Innsbruck, Austria; Gerold.Untergasser@i-med.ac.at; 4Experimental Oncogenomic Group, Tyrolean Cancer Research Institute, 6020 Innsbruck, Austria; 5Pathology Laboratory Obrist and Brunhuber, 6511 Zams, Austria; Peter.Obrist@tyrolpath.at; 6Clinical Pathology and Cytodiagnostics, tirol-kliniken, 6020 Innsbruck, Austria; Iris.Pipp@innpath.at; 7Department of Radiology, Medical University Innsbruck, 6020 Innsbruck, Austria; Fabian.Steinkohl@student.i-med.ac.at

**Keywords:** Lynch syndrome, urothelial cancer, upper urinary tract, DNA mismatch repair genes, MMR, microsatellite instability, immunotherapy, checkpoint inhibitor

## Abstract

Lynch syndrome, known as hereditary nonpolyposis colorectal cancer (HNPCC), is an autosomal-dominant familial cancer syndrome with an increased risk for urothelial cancer (UC). Mismatch repair (MMR) deficiency, due to pathogenic variants in *MLH1*, *MSH2*, *MSH6*, and *PMS2*, and microsatellite instability, are known for development of Lynch syndrome (LS) associated carcinogenesis. UC is the third most common cancer type in LS-associated tumors. The diversity of germline variants in the affected MMR genes and their following subsequent function loss might be responsible for the variation in cancer risk, suggesting an increased risk of developing UC in *MSH2* mutation carriers. In this review, we will focus on LS-associated UC of the upper urinary tract (UUT) and bladder, their germline profiles, and outcomes compared to sporadic UC, the impact of genetic testing, as well as urological follow-up strategies in LS. In addition, we present a case of metastatic LS-associated UC of the UUT and bladder, achieving complete response during checkpoint inhibition since more than 2 years.

## 1. Introduction

Lynch syndrome (LS), also known as hereditary nonpolyposis colorectal cancer (HNPCC), is the most common hereditary type of colon cancer, accounting for 3% of new colon cancer diagnoses [1]. Apart from early onset colon cancer with proximal predominance and an excess of synchronous and/or metachronous colon cancer [2], the cancer types most frequently associated with LS are endometrial and ovarian cancer [3], and malignancies affecting the stomach, small bowel, prostate, breast, brain, and hepato-biliary tract [4,5,6,7]. Due to the increased recognition of tumor heterogeneity in LS, LS-associated cancer types include urothelial cancer (UC) as the third most common cancer in subsets of these families as first described by Lynch et al. in 1990 [8]. Therefore, the aim of this review article is to focus on LS-associated UC of the upper urinary tract (UUT) and bladder, highlighting genetics, molecular subtype classifications, and germline variant profiles compared to sporadic UC, the impact of genetic testing, incidence, outcomes, and potential future treatment strategies such as immune checkpoint inhibitors (ICI), which might help to optimize treatment in LS-associated cancer.

### 1.1. Materials and Methods

Literature research was performed by search of the commonly used databases PubMed^®^ (National Library of Medicine, National Center for Biotechnology Information, Bethesda, MD, USA) and MedLine^®^ (Medical Literature Analysis and Retrieval System Online, National Center for Biotechnology Information, Bethesda, MD, USA), which were accessed until July 2020. The study language was limited to English language. The following medical subject heading terms were used to identify the results suitable to our review topic: Lynch syndrome, urothelial cancer, upper urinary tract, mismatch repair genes, microsatellite instability, immunotherapy, checkpoint inhibitor. A workflow, shown in Figure 1, gives on overview of our literature search structure.

Thereafter, suitable findings that we presumed to be of clinical interest were included in the detailed analysis in our review article.

### 1.2. Lynch Syndrome and Genetics

Sporadic colon cancer is often linked to point mutations in tumor suppressor genes such as *p53* [9] and *APC* [1], which are less represented in LS [2]. After the onset of MMR in LS, mutations in the KRAS gene commonly occur, followed by APC mutations [10,11]. The syndrome causes a wide variety of oncological cancer types. Its underlying mechanism is a germline variant of DNA mismatch repair (MMR) genes, which are found in 88% to 95% of patients diagnosed with this disorder [3,4]. Additionally, *EPCAM*, which is a gene adjacent to *MSH2* that when mutated can cause the *MSH2* to be inactivated, constitutes for about 3% of LS cases [12]. MMR is one of the essential factors in preventing cancer development in a biological system, as it corrects miss-paired DNA insertions and replication errors and serves as a checkpoint to maintain vital genomic stability by restoring improperly assembled single-base matches during replication [5]. In addition, oxidation [6] or alkylation damage can be recognized and either be repaired, or cell apoptosis can be induced. Germline variants in one of two MMR genes is a precondition in individuals with LS. Inactivation of the second healthy allele usually arises due to small pathogenic variants and gene loss, leading to a defect in the MMR system [13].

Four main MMR genes have been found to be responsible for LS associated cancer development to date. These predisposing genes are mutL homolog 1 (*MLH1*) [14] and mutS homolog 2 (*MSH2*) [15], followed by mutS homolog 6 (*MSH6*) [16] and postmeiotic segregation increased 2 (*PMS2*) [17]. *MSH2* has been shown to form the *MSH2-MSH6* heterodimer, identifying base-base mismatches. The *MSH2* and *MLH1* proteins are the mandatory partners of their particular dimer [18]. MutLα then promotes the excision of the mismatched locus, which is performed by proteins such as exonuclease-1 and DNA polymerase, and DNA ligase resynthesize and ligate the DNA strand [19,20]. The sequence of synthesis is illustrated in Figure 2.

Any defect in these proteins results in an inactive DNA repair process, which increases pathogenic alteration rates in genes of the cell growth cycle, leading to defects in tumor suppressor genes and oncogenes, followed by an elevated cancer risk [22]. Insufficient MMR and subsequent variation in the number of nucleotides in a microsatellite region, defined as small repetitive DNA sequences, is referred to as microsatellite instability (MSI), which is present in approximately 95% for of all tumors associated with LS [15]. In contrast, MSI is not common in sporadic colorectal cancer (CRC). Tumor mutational burden (TMB) was found to be consistently high across colorectal cancer with MMR gene inactivation [23]. Fabrizio et al., additionally reported that TMB assessment can accurately classify MSI tumors as TMB-high [24], as TMB is known to be a predictive response marker in ICI therapy [25,26], and it underlines the use of targeted therapy in patients with LS.

### 1.3. Lynch Syndrome and Cancer Susceptibility

Interestingly, current findings from the prospective LS database confirmed distinct gene and gender-specific patterns of cancer risk depending on the affected MMR pathogenic variant carrier in LS patients. In detail, *MSH2* pathogenic variant carriers confirmed a higher risk of UC of the UUT. UC of the bladder is most related to *MSH2* pathogenic alterations, yet not to the extent of UC of the UUT. Interestingly, neither urinary tract cancers, nor colorectal or endometrial cancer (EC), were observed before the age of 50 in carriers of a *PMS2* variant. At older age, it has been shown that germline pathological variants of *MSH2* and *MLH1* are even more associated with urinary tract cancer. In EC, MMR deficiency has already been shown in 20–40% of cases, with still controversial data on its prognostic value [27]. Additionally, the *MSH6* pathogenic variant seems to be a gender-specific factor in cancer susceptibility with higher EC risk, but only a low risk of colon cancer in both sexes [7].

The findings result in cumulative incidences at an age of 75 years for urinary tract cancers with *MLH1*, *MSH2*, and *MSH6* variants in 8%, 25%, and 11%, respectively [28], suggesting that *MSH2* pathogenic variant carriers were at significant increased risk of developing UC compared with individuals with pathogenic germline variants in *MLH1* or *MSH6*. Loss of co-expression of *MLH1* and *PMS2*, as described above, was shown to be associated with lower mutational burden (TMB) in contrast to loss of the *MSH2/MSH6* dimer. This variability may be responsible for tissue-specific differences and the variation of cancer risk in LS [29]. Additionally, it has been demonstrated that tumors with MSI due to defects in MMR proteins often present with high Th1 type T-cell infiltration [30,31,32]. Additionally, MMR protein deficient tumors characteristically have T cell infiltration that could indicate an ongoing antitumor immune response and may counteract the immunosuppressive effects of standard chemotherapy [33]. A detailed overview of incidence rates of LS-associated cancers at 75 years depending on the involved MMR gene mutation is presented in Figure 3.

### 1.4. Diagnosis of Lynch Syndrome

Several clinical models and strategies have been developed for identification of patients at high risk for LS. The first guideline criteria were published in 1990 as the Amsterdam criteria I, in which all criteria listed in Table 1 had to be fulfilled by affected families [34]:

As mentioned, LS includes various extra-colonic malignancies, which were not considered in these first diagnostic criteria. In 1999, the revised Amsterdam criteria II stipulated at least three relatives with LS-associated cancer instead of only CRC, defined as CRC, cancer of the endometrium, small bowel, ureter, or renal pelvis [35]. Because LS is a hereditary disease, anamnesis and an accurate medical family record concerning malignancies are essential for early screening and diagnosis, but in clinical practice this can be difficult [36]. Histopathology alone is not usually able to distinguish LS associated CRC from sporadic CRC cases [37], but specific histopathological characteristics have been described as being representative for LS. These include peritumoral Crohn’s like lymphatic reaction, medullary histology with poor differentiation, and intratumoral lymphocytic infiltration [38,39]. Mucinous histopathologic character is often present [40,41]. Identifying individuals who do not meet the Amsterdam criteria, yet have a potential risk of LS, is still a challenge. For this reason the Bethesda Guidelines were defined in 1997 which, when fulfilled, propose testing of replication errors and MSI to detect mutation carriers [42]. The Bethesda guidelines were revised in 2004 and are listed in Table 2 [43].

The MIPA criteria, published in 2005 and listed in Table 3, propose a simplified version of the Bethesda Guidelines by excluding details of family history [44], which are often not applicable in clinical practice [36].

The molecular diagnostic approach combines MSI analysis and MMR protein immunostaining and seems to be a productive way of pre-selecting those patients at high risk for LS for further germline variant analysis [45]. Ninety-five percent of LS-associated cancers are MSI-positive [46]. Tumors are characterized on the basis of frequency of instability: Low-frequency MSI (MSI-L) and high-frequency MSI (MSI-H), which is suspicious for LS. MSI-H status has been associated with better survival due to accumulation of somatic mutations [47,48,49]. In CRC, MSI is a well-known prognostic marker [50]. Furthermore, in patients with stage II disease of CRC, MSI is also of predictive importance for the response to immunotherapy in metastatic disease [51]. Association between MSI and IHC in extra-colonic LS cancers has been shown to become more and more weaker, suggesting that microsatellites are organ-specific [50]. Different staining patterns in immunohistochemistry (IHC) allow an assessment of MMR protein loss and have been proposed as an alternative screening method to MSI testing, both being valid and effective diagnostic tools for selecting patients for further genetic germline variant analysis [52,53]. However, IHC cannot differentiate between dysfunctional proteins deriving from either missense mutations or polypeptides. It has been shown that *MSH2* and *MLH1* proteins are the obligatory partners of their heterodimers *MSH6* and *PMS2*, respectively (Figure 1). Furthermore, *MSH2* and *MLH1* are stable without their dimer partners, but not reversely. To conclude, tumors with an *MLH1* pathogenic germline variant will show loss of both partners *MLH1* and *PMS2.* Pathogenic variants in *MSH2* will result in disintegration to missense of both *MSH2* and *MSH6*. On the other hand, pathogenic variants in the secondary genes *MSH6* and *PMS2* result in selective loss of only these genes, as additionally summarized in Table 4. IHC should therefore include antibodies of all four proteins in order to detect as many as *MLH1* and *MSH2* abnormalities as possible. Thus, various IHC expression patterns associated with MMR-gene pathogenic variants are possibly based on the heterodimeric nature of the MMR proteins as already described by Van Lier et al. [45].

In comparison, IHC may be slightly superior in predicting pathogenic germline variants to MSI analysis. Additionally, it is cheaper and more easily applicable [54] as a fast and simple procedure. IHC for MMR proteins should not replace MSI testing to detect LS, encouraging a so called ‘combined diagnostic molecular concept’, as IHC interpretation may be difficult due to inter- and intraobserver variability, missense mutations, or low or absent intensity of nuclear staining in tumors and normal tissue [55,56]. Nevertheless, both molecular screening methods, IHC staining and MSI analysis, confirm the absence of one the four MMR proteins or a high MSI count, which assigns patients to further germline analysis. On the other hand, if the clinical suspicion for LS is very high, even in the case of no tumoral MMR deficiency or MSS tumors, genetic counselling and testing are still indicated, although the false-negative rate of MSI analysis is very low (<5%). Due to the fact that germline analysis is time consuming and expensive, the current standard practice is to select patients for genetic testing by a molecular diagnostic work-up, guided by clinical and pathological criteria, such as MSI testing and IHC staining of MMR proteins [45].

Genetic testing is performed on DNA isolated from peripheral blood mononuclear cells obtained by blood draw or from an oral rinse. Currently, in addition to *MLH1* and *MSH2*, whose testing was introduced in 1990s, mutations in the genes *MSH6*, *PMS2*, and *EPCAM* are now also included in testing of individuals suspected to have LS [57,58,59]. Germline testing confirms LS in only 24 to 67% of MMR-deficient CRC and in 16 to 80% of MMR-deficient EC [60,61], depending on the IHC expression patterns. Accordingly, germline analysis by next-generation sequencing has improved, generating a more comprehensive genetic profile of both germline and somatic mutations, thereby analyzing DNA isolated from both blood and tumor samples in parallel [62]. Subsequently, ‘paired tumor and germline testing’ seems to be an attractive diagnostic tool in the LS testing algorithm, as it identifies an underlying cause for MMR-deficient tumors such as CRC and EC in 76% and 61% of patients without and with prior LS germline testing. Combined testing allows the simultaneous diagnosis of LS and describes atypical IHC patterns in patients where germline pathogenic variants were not concordant with specific protein expression, e.g., double somatic mutations, which explain IHC staining in addition to a germline mutation in a different LS gene. Moreover, solely tumor sequencing for germline screening is limited in detecting exon-level somatic copy number variants in regions with significant pseudogene homology such as the *PMS2* pseudogene region [63]. These findings may corroborate the importance of performing a comprehensive germline testing regardless of IHC staining patterns, reducing the risk of germline rearrangements.

### 1.5. Lynch Syndrome and UC of the UUT and Bladder

UC of the UUT accounts for only 5 to 10% of UC [64], but about 60% of primary diagnosed UUT cancers are invasive, and 7% have already metastasized at first diagnosis [65,66]. Tumors of the renal pelvis are 3 to 4 times more common than malignancies of the ureteric tract [67]. Hereditary UC of the UUT is linked to LS and accounts for approximately 20% of all UC of the UUT [68,69], and it ranks third (5%) after colon cancer (63%) and EC (9%) within the group of HNPCC-associated tumors [70]. Patients diagnosed with LS have a 22-fold higher risk and a cumulative lifetime risk of 3% to develop UC of the UUT, as compared to the general population [70,71]. The relative risk of UUT cancer in patients with LS therefore is 14% [72]. Compared to sporadic UC, LS-associated UC is known to have a female predominance and to occur at a younger age [73]. Hubosky et al. first described that patients with LS who develop UC of the UUT in their cohort appear to be more likely to have bilateral UC of the UUT over their lifetimes compared to sporadic UUT cancer patients [25].

Whereas UC of the UUT and bladder are biologically and histomorphologically very similar [74], differences in germline variant profiles have been noticed, especially in high-grade urothelial tumors [75]. Comparison of next-generation sequencing of high-grade UUT cancer with high-grade UC of the bladder identified similar pathogenic germline variants in both cancer types, but at different frequencies [76], confirming a higher prevalence of fibroblast growth factor receptor 3 (*FGFR3*), *HRAS* and *CDKN2B* mutated genes in high-grade UUT cancers, and *TP53* and *RB1* in high-grade UC of the bladder [76,77]. *FGFR3* is a member of the tyrosine kinase family and its signaling plays a role toward cell differentiation and proliferation [78] and is known to contribute to an early and essential stage in the molecular pathogenesis of papillary bladder cancer [79,80]. Moss et al. confirmed four unique molecular and clinical subtypes of UC of the UUT by whole-exome sequencing of DNA, RNA sequencing, and protein analysis, as described in Table 5.

A systemic evaluation of current genomic sequencing and proteomic data in UC of the UUT confirmed molecular differences in *FGFR3* (fibroblast growth receptor 3), *TP53*, and MSI between sporadic (80–90%) and hereditary (10–20%) UUT cancers. Whereas sporadic low-grade UC of the UUT is *FGFR3* mutated in over 90%, high-grade UC of the UUT is associated with *TP53/MDM2* mutations. Nevertheless, UC in patients with LS share molecular similarities of subtype classification with sporadic UC, identifying a predominance of the ‘urothelial-like’ molecular subtype in LS with only the remaining 20% being genomically unstable, basal-like, or related to other subtypes [73]. MMR protein loss has been shown to be present in 7% of all UUT cancer cases and 30% in LS-related UC of the UUT, in this cohort of patients with UUT urothelial cancers and verified loss of mismatch repair protein expression, up to 86% were affected in loss of *MSH2* and *MSH6*, with the remaining 14% showing isolated loss of *MSH6* [82]. Cases with *MSH2* variants were shown to have the highest risk of developing UC [73,83,84], with an odds ratio of 4.6 (*p* = 0.001) [85]. Development of UC has been shown to be strongly associated with *MSH2* variants in up to 73%, leaving patients with this specific protein-loss at higher risk for UC than individuals with loss of *MLH1* or *MSH6*. [15]. MSI is described to be rare in bladder cancer and to be present in up to 40% of cases of UC of the UUT cases [86,87]. Certain clinical and pathological criteria are more common in UC of the UUT with MSI, such as location in the lower ureter, female sex, younger age, and an inverted growth pattern [88].

Huang et al. have reviewed the association between LS and bladder cancer. It has to be said that it is difficult to assess whether bladder cancer is a LS-associated malignancy, as data regarding the risk of bladder cancer in LS are sparse, yet penetrance was shown to be much less than for LS-associated UC of the UUT [89]. Generally, it has been shown that bladder cancer generally was more common in *MSH2* variant families than in the general population. Additionally, these patients had evidence of UC of the UUT as well, which leads the authors to assume that individuals with bladder cancer had an increased risk for UC of the UUT in the first place [90]. Especially amongst patients with *MSH2* pathogenic variants, the incidence not only for UC of the UUT, but also for bladder cancer, has shown to be increased [91]. A Swedish cohort of patients fulfilling the Bethesda criteria showed an increased risk of malignancies of UC of the UUT, but interestingly not of UC of the bladder itself [92]. In contrast, van der Post et al. confirmed an increased risk of UC of both the UUT and the bladder in patients with LS carrying a germline *MSH2* variant. The cumulative risk of bladder cancer alone until the age of 70 years in *MSH2* pathogenic variant carriers and first-degree relatives was 12.3% for men and 2.6% for women. The overall cumulative risk for urinary tract cancer, including bladder and UUT, in *MSH2* germline variant carriers and first-degree relatives was 18.2% in men and 8.4% in women [84]. However, there are several factors that may point to the link between bladder cancer and LS [89]. First of all, patients with prior or concurrent UC of the UUT can confound results. Moreover, the low rate of MSI in UC of the bladder may be due to the association with LS [93]. Most importantly, patients with LS-associated UC have similar overall survival rates compared to stage-matched sporadic UC, but are diagnosed at younger age in female patients. In addition, in MMR-deficient LS patients the overall survival was not significantly influenced by MSI [73]. In patients who underwent nephroureterectomy due to invasive UC of the UUT, a high MSI incidence (17%) was an independent positive prognostic factor for survival, especially in patients younger than 71 years with tumor stage T2-T3N0M0 [94]. Five-year and 10-year survival rates for LS-associated bladder cancer with germline pathogenic variants in any of the four MMR genes were very promising with 93% and 81%, respectively [28]. These data suggest that bladder cancer is part of the LS tumors spectrum and consequently, surveillance should be considered, especially in *MSH2* pathological germline variant carriers.

### 1.6. Recommendations of Screening in LS

Early diagnosis of LS enables precise screening for potential LS-associated tumors and inclusion into follow-up programs, reducing the lifetime cancer risk [95,96]. Universal molecular testing can miss up to 28% of LS cases after applying the revised Bethesda Guidelines [95]. Specific screening of patients presenting with UC of the UUT using IHC of MMR proteins and the revised Amsterdam Criteria (ACII) [35] showed that in patients with no known cancer predisposition, up to 21% were estimated to have underlying LS [97]. The sensitivity and specificity of immunohistochemistry are approximately 83% and 89% [95,98]. Furthermore, specific histopathological features, such as pleomorphism, inverted growth, and intratumoral lymphocytes, were associated with the presence of LS [99]. MSI occurred significantly more often when the pathological feature of inverted growth was present, leading to the consideration that this histological parameter could serve as a marker lesion for MSI and help identify patients with LS with UC of the UUT. Using positive ACI/II, MSI, or IHC, Metcalfe et al. identified 13.9% of UC of the UUT as potential LS-tumors. Subsequent genetic testing confirmed LS in 5.2% [97], which is still highly frequented and highlights the importance of this genetic syndrome in UC. Screening for EC and CRC for loss of MMR proteins has already become a standard of care [100,101]. Usage of IHC and PCR was implemented for screening of all new CRC cases, resulting in 94% concordance [102,103]. Similar screening concepts, which are clinically easy applicable and cost-effective, are yet to be established to identify patients with LS-associated UTUC. Currently, the only recommendation for testing of LS is by the European Association of Urology (EAU) Guidelines. Patients at high risk for HNPCC syndrome, which means age <60 years, a personal history of HNPCC-related cancers or first-degree relative <50 years with HNPCC-related cancers, or two first-degree relatives with HNPCC-related cancers [68] should undergo germ-line DNA sequencing with family counselling [64]. In a cohort of 117 patients, it was demonstrated that previously suggested demographic and histologic factors were found to be ineffective at flagging patients with UC of the UUT for further screening, proposing reflexive MMR screening by IHC followed by MSI testing for all UC of the UUT patients [104]. Implementation of a universal screening method remains challenging. European guidelines for managing patients with MMR mutations recommend sonography and urine analysis every one to two years from the age of 30 only in family constellations of two or more UC of the UUT cases [105].

### 1.7. Recommendations of Urological Surveillance in LS

Mork et al. suggested surveillance of patients with diagnosed LS, yet without development of UC of the UUT. The recommendations include (1) frequent urinalysis with a threshold of three red blood cells per high power field for further investigation, (2) CT scan including a urographic phase when follow-up of CRC is performed, and (3) cystoscopy with a retrograde pyelography. Urine cytology is not recommended [106]. There is general agreement that regular urinalysis should be the leading diagnostic examination when it comes to surveillance in patients diagnosed or suspicious for LS [70,107,108]. In cases of UC of the UUT, MSI testing therefore should be sought, and hereditary predisposition should be investigated, pursuing patients with *MSH2* protein loss in IHC to undergo further testing for germline mutations. Special attention should be put on surveillance of UC of the bladder in patients with a verified *MSH2* variant, as the association between *MSH2* and UC has evolved in the past [15,109,110]. In literature, recommendations for UC surveillance in LS are defined as ultrasound of the bladder and UUT with urine cytology and sediment in every *MSH2* mutation carrier starting at age 40 and above, performed every one to two years [84].

### 1.8. Lynch Syndrome and Immunotherapy

Immune checkpoint inhibitors (ICI) have had a profound impact on the oncological treatment landscape and are known to act as regulators of T cell activation [111]. The core concept is the ability of ICI to enhance the anti-tumor activity of infiltrating T cells. The immune checkpoints (IC) targeted by ICI are programmed cell death 1 (PD-1), programmed cell death ligand 1 (PD-L1), and cytotoxic T lymphocyte antigen 4 (CTLA-4). IC prevent infiltrating T cells from attacking tumor cells. Conversely, blockade of these IC by ICI abrogates the immunosuppressive effects and facilitates antitumor T cell responses. MMR protein loss causes an upregulation of mutation-associated neoantigens, which then trigger more T cells to infiltrate the tumor [112]. However, PD-1 and PD-L1 are often upregulated in infiltrating T cells, preventing antitumor responses. These tumors are reported to express high PD-(L)1 levels on their cell surface [113,114]. Upregulated expression of immune checkpoints, e.g., PD-L1, then leads to inhibition of cytotoxic T cell activity in the tumor environment and thereby promotes immunosuppression, associated with poorer overall survival [115]. Administration of PD-1 and PD-L1 inhibitors in MSI tumors can therefore be held accountable for good response rates. Cancers with proven MSI produce large amounts of neo-antigens that tumor infiltrating lymphocytes (TILs) subsequently target; these tumors are reported to express high PD-(L)1 levels on their cell surface [113,114]. In the advanced therapy-refractory treatment setting, it has been shown that MSI tumors are suitable for immunotherapy [116]. In the Checkmate 142 phase 2 trial, nivolumab achieved further durable responses, consisting of an objective response rate (ORR) of 31%, with almost 69% of patients having disease control for longer than 12 months [117]. Previous case reports have also shown a better response in both CRC and UTUC in patients who had received pembrolizumab after failure of chemotherapy [118,119]. Based on the results of the Checkmate 142 trail, nivolumab received approval for metastatic colorectal cancers with MMR-deficiency and MSI-H that has progressed following prior treatment with fluoropyrimidine, oxaliplatin, and irinotecan [117]. Le et al. found that patients with MMR deficiency showed a significantly higher ORR (40%) compared to patients with MMR proficiency (11%) in colon cancer in response to pembrolizumab treatment. In this cohort, MSI was a significant predictor of ORR [32]. This was also verified for tumors with MMR deficiency regardless of their origin [120], which finally resulted in FDA approval of pembrolizumab in MMR deficient tumors [121]. In epithelial ovarian cancer, the common histologic subtype of high grade serous cancer showed higher PD-L1 positivity and a higher rate of TILs, supporting the consideration of immunotherapy being a promising treatment strategy for this type of ovarian cancer [122]. Recently, pembrolizumab has been approved as first line therapy of metastatic MMR/MSI colorectal cancer [123]. Currently, trials that use ICI in LS-associated tumors are restricted to CRC with no ongoing trials in LS-associated UC. Yet, they point out the upcoming importance of immunotherapy in all MMR deficient tumors. A detailed list of studies concerning LS is shown in Table 6.

## 2. Case Presentation

In the following, we present the case of a 53-year-old patient who developed UC of the right renal pelvis who, at the age of 48, has been diagnosed with an adenocarcinoma of the stomach, managed by gastrectomy and perioperative chemotherapy with 5-fluorouracil, folic acid, oxaliplatin, and docetaxel. Gastrectomy specimens showed histology of adenocarcinoma of the stomach, described as intestinal type according to Lauren’s criteria, staged ypT3 with resection boundaries free of cancer cells (R0). Histopathology also showed evidence of MSI and loss of expression in the mismatch repair gene *MSH2*, as shown in Figure 4.

The patient was a non-smoker, and the family history revealed that his aunt had developed CRC at the age of 84, his brother was diagnosed with an oncological disease of unknown entity, and there was a cousin marriage between his parents. At follow-up, nine months after gastrectomy, a computed tomography (CT) scan showed a suspicious contrast defect in the right renal pelvis (Figure 5A). After undergoing ureterorenoscopy, urothelial cancer staged pT1, grade 2 was confirmed in the biopsy material. The patient then had laparoscopic nephroureterectomy at our institution with the pathological finding of high-grade UC of the UUT, staged pT3 with resection boundaries free of cancer cells (R0). Six months later, the patient developed intermittent microhematuria and mucosal alterations at the ostium of the right ureter at cystoscopy on follow-up. A CT scan additionally showed the presence of a contrast medium enhancing mass in the right ostial region of the bladder (Figure 5B) and enlarged retroperitoneal lymph nodes. Transurethral resection of the bladder showed the finding of current urothelial cancer, staged pT2a, where after two cycles of neoadjuvant cisplatin-based chemotherapy (gemcitabine 1000 mg/m^2^ on days 1, 8, and 15, and cisplatin 70 mg/m^2^ on day 2; 1 cycle = 28 days) were administered [124] followed by radical cystoprostatectomy (RC) with ileal conduit and extended pelvic lymphadenectomy. Final histology at RC confirmed pathological complete response, staged ypT0R0L0V0N0.

Three months later, enlarged retroperitoneal lymph nodes were found during oncologic follow-up (Figure 5C). These were biopsied under CT guidance (Figure 5D) and revealed a metastasis of the known UC of the bladder. After meeting the clinical criteria with the presence of metachronous LS-associated tumors, MSI and loss of expression of the MMR protein *MSH2* was confirmed. With the knowledge that *MSH6* expression can be retained in absence of *MSH2* staining [125] and given the clinical suspicion for LS, this patient was assigned to germline testing—confirming a germline *MSH2* mutation, and therefore LS [95]. IHC of both gastrectomy and cystectomy specimens showed loss of *MSH2* expression. (Figure 4 and Figure 6).

Therapy with pembrolizumab was started at a dose of 200 mg every three weeks. Three months later, after five cycles, CT imaging showed partial remission, with a significant decrease of the retroperitoneal mass (Figure 5E). After an additional 30 cycles of pembrolizumab, the patient showed complete remission (Figure 5F). Therapy was very well tolerated, and the patient is in good general health since more than 2 years. In accordance with previous observations, our case report shows immunotherapy to be suitable and efficient in patients with LS or MMR pathological germline variants.

## 3. Conclusions

Hereditary urological cancers, especially of the UUT, can often be misclassified as sporadic, resulting in an underestimation of their incidence. Hereditary UC of the UUT accounts to approximately 20% of all UC of the UUT cases and ranks third (5%) after colon cancer (63%) and EC (9%) within all reported LS-associated tumors. Compared to sporadic UC, LS-associated UC, especially in the UUT, is associated with a female predominance, bilateral occurrence, and younger age at first diagnosis. Specific histopathological features, such as intratumoral lymphocytes and inverted papilloma-like growth pattern, may be useful in combination with IHC in identifying those at risk for LS. Urological surveillance in LS should include imaging of the UUT, urine analysis, cytology, and cystoscopy, especially in patients with *MSH2* germline variants. MSI was detected more frequently when the pathological feature of inverted growth was present, leading to the consideration that this pathological feature could serve as a surrogate marker for MSI and help identify patients with LS and UC of the UUT at first oncological diagnosis. In cases of UC of the UUT, MSI testing should be sought, and hereditary predisposition should be investigated by genetic testing. In this context, IHC serves as a clinically more applicable strategy to identify patients with MMR protein loss for subsequent genetic testing. Since MSI, which occurs in up to 40% of UC of the UUT cases, is associated with a better survival, and given that tumors with a high MSI show good responses to immunotherapy, it is tempting to predict that ICI will have a central role in the therapeutic regimes of LS.

As presented in our case, ICI show promising results in the therapeutic landscape of LS-associated tumors, which is supported by our case presentation. Future research is dedicated to the development of guidelines that provide a framework for screening, surveillance, and therapy regimes of patients diagnosed with LS-associated UC.

## Figures and Tables

**Figure 1 ijms-22-00531-f001:**
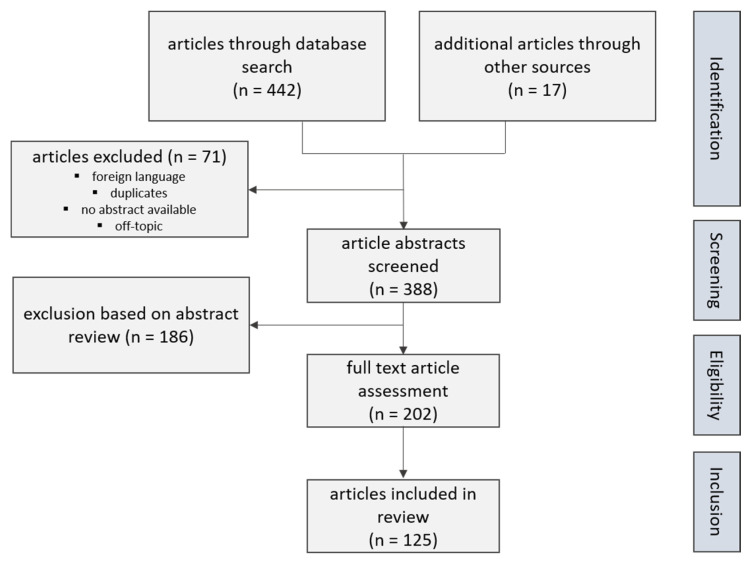
Workflow giving an overview of our literature search structure.

**Figure 2 ijms-22-00531-f002:**
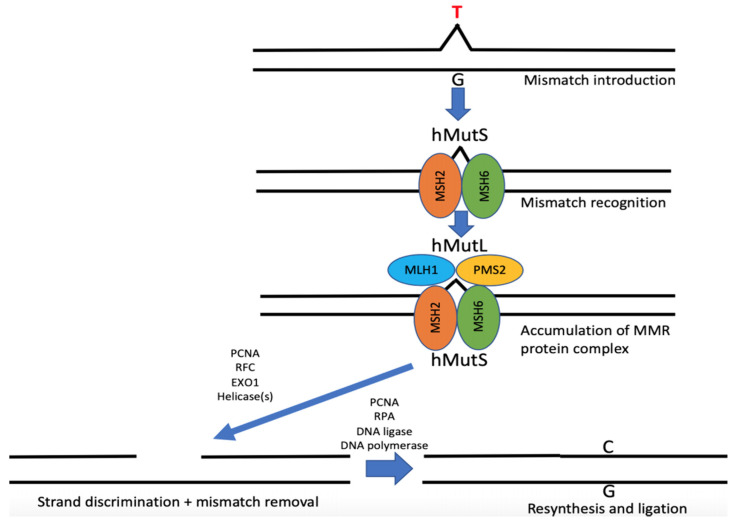
Sequence of protein synthesis. Four DNA mismatch repair genes are responsible for Lynch syndrome-related cancer development. *MSH2* could bind mismatched nucleotides together with *MSH6*. *MLH1* complexes with *PMS2,* thus forming the MutLα complex, which is responsible for excision of the mismatched locus. Defects in these proteins may increase the percentage of mutations and diminish effectiveness of tumor suppressors. Adapted from Peltomäk et al. [21].

**Figure 3 ijms-22-00531-f003:**
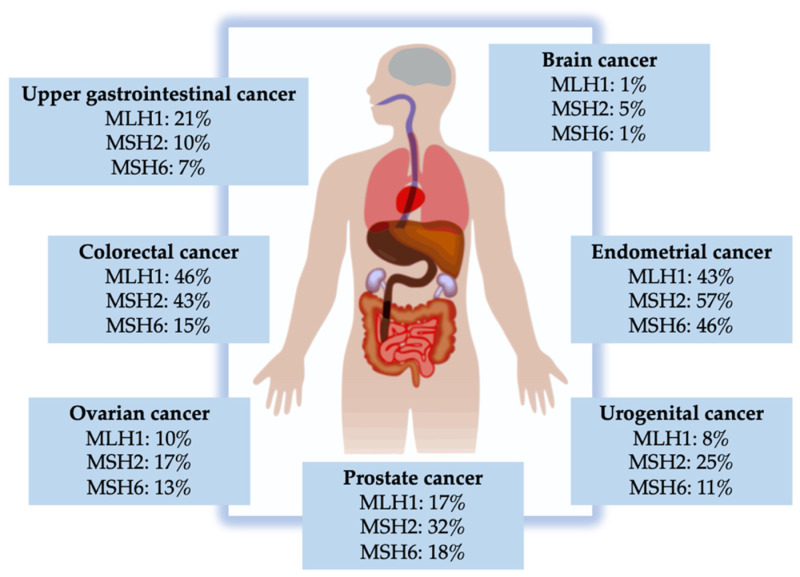
Incidence rate of Lynch syndrome (LS)-associated cancers depending on the involved mismatch repair (MMR) gene mutation up to the age of 75 years [28].

**Figure 4 ijms-22-00531-f004:**
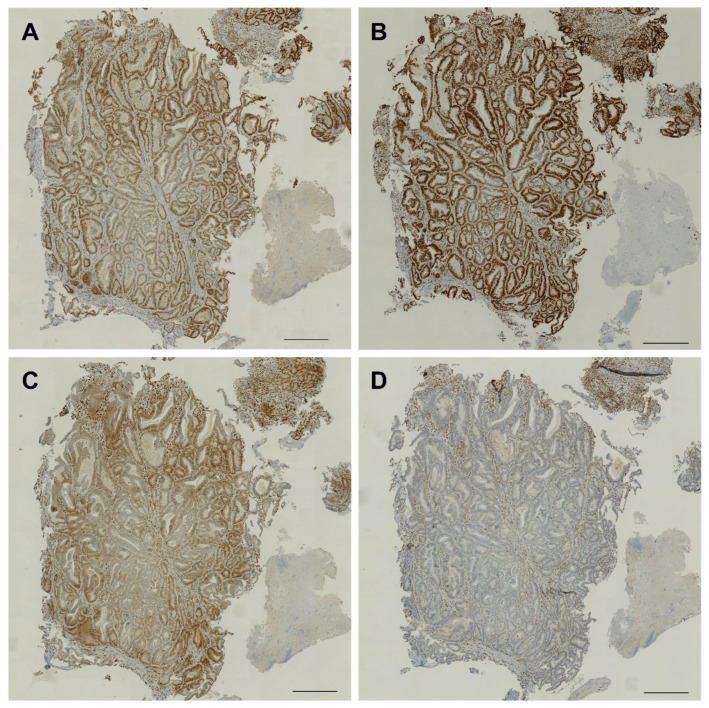
Immunohistochemical analysis of mismatch repair protein complexes in gastric biopsy. Gastric mucosa staining positive for *MLH1* (**A**), *PMS2* (**B**), and *MSH6* (**C**), and lost expression of *MSH2* (**D**). Scale bar indicates 100 µm.

**Figure 5 ijms-22-00531-f005:**
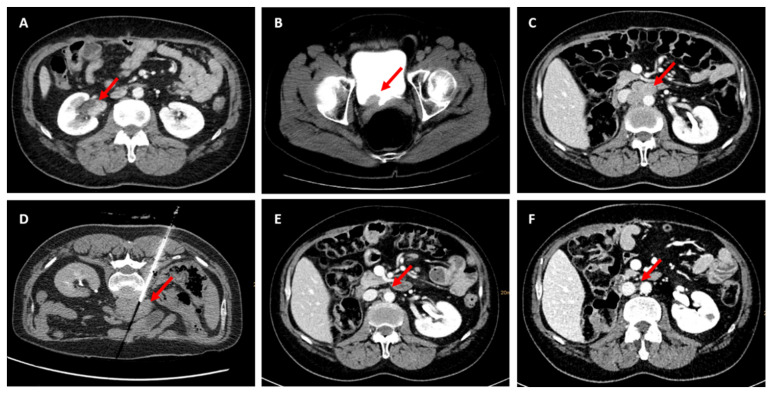
Frequent computed tomography (CT) scans of a patient diagnosed with LS during intravenously administered therapy with pembrolizumab. (**A**) Axial CT scan with an enlarged mass of the right renal pelvis (red arrow) in first diagnosis. (**B**) Axial CT scan prior to electroresection of the bladder, showing an enhancing mass (red arrow) at the region of the right ureter ostium. (**C**) Retroperitoneal lymph node bulk (red arrow) occurring three months after cystoprostatectomy. (**D**) CT-guided biopsy of the interaortocaval lymph nodes (red arrow). (**E**) Normal sized lymph nodes (red arrow) after five cycles of pembrolizumab 200 mg. (**F**) Remaining normal sized lymph nodes (red arrow) after 28 cycles pembrolizumab 200 mg with no other signs of regional or local metastasis.

**Figure 6 ijms-22-00531-f006:**
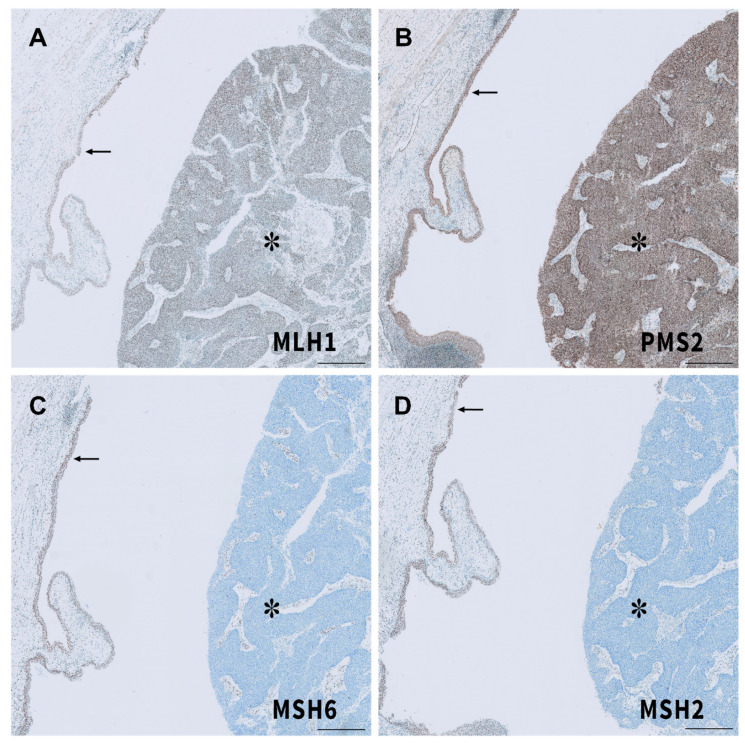
Immunohistochemical analysis of the mismatch repair protein complexes in the radical cystectomy specimen: Urothelial mucosa staining positive for *MLH1*, *PMS2*, *MSH6*, and *MSH2* (arrow in (**A**–**D**)) versus lost expression of *MSH6* (**C**) and *MSH2* (**D**) in the urothelial carcinoma (asterisk).

**Table 1 ijms-22-00531-t001:** Amsterdam criteria I for LS patient identification [34].

Amsterdam Criteria I
There should be at least three relatives with a CRC (colorectal cancer).
One should be a first-degree relative of the other two.
At least two successive generations should be affected.
At least one should be diagnosed before the age of 50 years.
Familial adenomatous polyposis (FAP) should be excluded.
Tumors should be verified by pathological examination.

**Table 2 ijms-22-00531-t002:** Bethesda Guidelines as current recommendation for LS identification.

Revised Bethesda Guidelines
Patients meeting any one of the following should undergo microsatellite instability (MSI) testing:
CRC diagnosed in an individual under age 50 years.
Presence of synchronous, metachronous colorectal, or other LS-associated tumors *, regardless of age.
CRC with the MSI-H (high-frequency MSI) histology ^‡^, in a patient <60 years of age.
CRC diagnosed in 2 or more first- or second-degree relatives with LS-related tumors *, regardless of age.
CRC in 1 or more first-degree relatives with a LS-related tumor *, with 1 of the cancers being diagnosed under age 50 years.

* Endometrial, ovarian, gastric, small bowel, pancreas, hepatobiliary tract, renal pelvis, or ureter and brain tumors, sebaceous gland adenomas, and keratoacanthomas. ^‡^ Presence of tumor-infiltrating lymphocytes, Crohn’s like lymphocytic reaction, mucinous differentiation, or medullary growth pattern [25,26].

**Table 3 ijms-22-00531-t003:** MIPA criteria. Identifying LS in patients without known family history.

MIPA Criteria
Patients meeting any one of the following should undergo MSI analysis:
CRC before the age of 50 years.
Two LS-associated tumors, including synchronous or metachronous CRCs or LS-associated tumors.
Adenoma before the age of 40 years.

**Table 4 ijms-22-00531-t004:** Protein expression in MMR-gene mutations.

Gene	MLH1	MSH2	MSH6	PMS2
MLH	− *	+	+	+
MSH2	+	−	+	+
MSH6	+	−	−	+
PMS2	−	+	+	−

** MLH1* gene mutation or promoter hypermethylation. + Positive nuclear protein expression in tumor and normal cells. − Negative nuclear protein expression in tumor cells and positive staining in normal cells.

**Table 5 ijms-22-00531-t005:** Proposal of four clinical and molecular subtypes of urothelial cancer (UC) of the upper urinary tract (UUT) [81].

**Cluster 1**	No *PIK3CA* mutation, non-smokers, high-grade < pT2 tumors, high recurrence
**Cluster 2**	100% *FGFR3* mutation, tobacco use, low-grade tumors, non-invasive disease, no bladder recurrences
**Cluster 3**	100% *FGFR3* mutations, 71% *PIK3CA*, no TP53 mutations, tobacco use, tumors all <pT2, five bladder recurrences
**Cluster 4**	*KMT2D* (62.5%), *FGFR3* (50%), *TP53* (50%) mutations, no *PIK3CA* mutations, tobacco use, high-grade pT2+ disease, carcinoma in situ, shorter survival

**Table 6 ijms-22-00531-t006:** Trials with immunotherapy in Lynch syndrome.

Agents	Targets	Comparator	Study	Study Phase	Status	Patient Enrollment	Study Number	Primary Outcome Measures	Secondary Outcome Measures
Pembrolizumab	PD-1	-	MK-3475-016	II	completed	113	NCT01876511	irPFS 20 w irORR PFS 20 w	OS irPFS 28 w ORR AE PFS 28 w DCR MSI as marker
Nivolumab/ Nivolumab + Ipilimumab/ Nivolumab + Ipilimumab + Cobimetinib/ Nivolumab + Daratumumab	PD-1 CTLA-4 MEK CD38	-	Checkmate 142	II	active, not recruiting	340	NCT02060188	ORR	ORR
Combination Chemotherapy + Atezolizumab	PD-L1	Combination Chemotherapy	NCI-2016-01417	III	recruiting	700	NCT02912559	DFS	OS AE
Pembrolizumab	PD-1	Standard of Care	Keynote-177	III	active, not recruiting	308	NCT02563002	PFS OS	ORR
Nivolumab	PD-1	-	NCI-2018-01491	II	active, not recruiting	3	NCT03631641	Adenoma incidence	-

irPFS—immune related progression free survival, irORR—immune related objective response rate, ORR—objective response rate, PFS—progression free survival, OS—overall survival, AE—adverse events, DCR—disease control rate, DFS—disease free survival.

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
