# Peer review of "Lynch Syndrome: Its Impact on Urothelial Carcinoma"

_ijms, 2021, doi:10.3390/ijms22020531_

Round 1

Reviewer 1 Report

Lidner et al., in their article entitled "Lynch Syndrome: Its Impact on Urothelial Carcinoma” make a comprehensive examination of urothelial carcinoma in the context of Lynch syndrome (LS), focusing on genetics, molecular subtype classifications and germline variant profiles as well as the impact of genetic testing, incidence, outcomes, and treatment strategies. In addition, a case of metastatic urothelial carcinoma associated with LS is presented to illustrate the new treatment approach.

The article is well structured, easy to read, and the selected references provide a complete and updated overview of the topic.

However, some changes should be made before publication.

MINOR CHANGES

1.-  The use of the abbreviations "UUT" for upper urothelial tract and "UTUC" for upper urothelial tract carcinoma should be more similar to facilitate understanding.

2.- In section 1.5 UUT Lynch and UC Syndrome and the Bladder, it would be best to use the number symbol (ex. 5) instead of the word (ex. five) in rows 207 and 209 for ease of reading.

3.- Clear the space between the number and the % symbol.

4.- Table 4. Remove the italics from the proteins. Consider simplifying the text in bold to avoid repeating the table description and adding "Gene" as a title in the y-axis.

5.- Table 5. Remove the bold in the explanation of Group 1. Avoid using a reference number (79) in the text of the table.

6.- The sentence on page 7, rows 234-236 is not clear. What does FGFR3 mean in parentheses?

7.- In section 1.6 Recommendations for screening and urological surveillance in LS, separate the two topics in different paragraphs.

8.- Figure 2. Improve image quality. It is not clear whether the tumor glands in MSH6 staining are positive or negative. The discrepancy between the lack of MSH2 staining and the retained nuclear expression in MSH6 staining should be explained.

9.- Figures 2 and 4 should be more similar in composition and size because they both illustrate the same in different tumors.

10.- Use the past tense in all the text of case report.

11.- Rewrite the sentence on page 11 rows 406-409, because loss of expression of MMR and MSI do not confirm a LS.

Author Response

Response to review comments

Manuscript ijms-1048443: „Lynch Syndrome: Its Impact on Urothelial Carcinoma

Dear Editors-in-Chief,

Dear Reviewers,

 thank you very much for giving us the opportunity to improve our paper according to the very constructive and interesting points raised by the two reviewers. We do appreciate the time and effort that the reviewers dedicated to providing feedback on our manuscript and are grateful for the insightful comments on and valuable improvements to our paper.

We have carefully read all comments and revised the manuscript as per reviewers’ suggestions, which we hope to meet with acceptance requirements. All changes performed in the manuscript have been highlighted in red letters to ease any further review process.        We included the reviewer comments immediately after this letter and responded to them individually, indicating exactly how we addressed each concern or problem and describing the changes we have made. 

We hope that after conscientious revisions our review is now suitable for publication and would appreciate      if our manuscript would be accepted for publication in the IJMS.

Yours sincerely,

Renate Pichler, MD, PhD, FEBU (corresponding author)

Review answers of the manuscript ijms-1048443

Comments reviewer #1

  1. The use of the abbreviations "UUT" for upper urothelial tract and "UTUC" for upper urothelial tract carcinoma should be more similar to facilitate understanding.

Answer: Dear Reviewer, we thank you for pointing out this necessary detail concerning abbreviation and wording. We critically went through the manuscript and now consistently used the term UC of the UUT, which is additionally marked in red throughout the review.

  1. In section 1.5 UUT Lynch and UC Syndrome and the Bladder, it would be best to use the number symbol (ex. 5) instead of the word (ex. five) in rows 207 and 209 for ease of reading.

Answer: Thank you for pointing out this little but important detail. We replaced the written-out number through its number symbol in the mentioned lines and marked them in red.

  1. Clear the space between the number and the % symbol.

Answer: Many thanks for mentioning this formatting issue. We removed the space in front of the percent symbol throughout the manuscript - for a better overview and reading flow of the revised manuscript, we waived the red markings after adjustment.

  1. Table 4. Remove the italics from the proteins. Consider simplifying the text in bold to avoid repeating the table description and adding "Gene" as a title in the y-axis.

Answer: We thank the precise reviewer for the comment on appearance of our table - we removed the upper first line with the former repetition of the table description. Furthermore, we removed the italics from the protein names and added the shared term “Gene”, as proposed, in the upper top left line.

  1. Table 5. Remove the bold in the explanation of Group 1. Avoid using a reference number (79) in the text of the table.

Answer: Thank you for commenting on this formatting issue - to adjust and implement your proposal, we removed the bold formatting of the definition of Cluster 1 and moved the reference number 81 into the table title.

  1. The sentence on page 7, rows 234-236 is not clear. What does FGFR3 mean in parentheses?

Answer: We thank the reviewer for this attentive notice and admit that this is a remaining sentence structure mistake from when drafting the review. We removed the parentheses and embedded the abbreviation into the sentence. The abbreviation ‘fibroblast growth receptor 3’ is already mentioned and defined above in line 261 and listed in the abbreviation list at the end of the manuscript.

  1. In section 1.6 Recommendations for screening and urological surveillance in LS, separate the two topics in different paragraphs.

Answer: Clear-cut structures are important for comprehension and reading flow, especially in a review manuscript - therefore many thanks for this advice. We streamlined and subdivided the section and deleted the paragraph in line 335 to merge the topic on screening and added a paragraph in line 352 to open up the following issue on surveillance textual, as well as visually.

  1. Figure 2. Improve image quality. It is not clear whether the tumor glands in MSH6 staining are positive or negative. The discrepancy between the lack of MSH2 staining and the retained nuclear expression in MSH6 staining should be explained.

Answer: Thank you for highlighting this issue and for giving this helpful advice to enhance our image quality. We rearranged and enlarged each specimen image to provide a high-resolution quality and to ensure a better identification of the positive tumor gland straining in the MSH6 analysis. The patient specific discrepancy in IHC staining of MSH2 and MSH6 is now explained and thereby gone into detail after streamlining the sentences in line 448 to 453, underlined by reference number 126. Additionally, former Figure 2 has been renumbered to Figure 4 due to two figure insertions in the above manuscript.

  1. Figures 2 and 4 should be more similar in composition and size because they both illustrate the same in different tumors.

Answer: We thank the reviewer for highlighting this detail concerning appearance of our newly numbered Figures 4 and 6 (due to insertion of Figure 1 and 3 in the above manuscript, as mentioned further below). Needless to say, clean-cut and identically arranged images round off a review in a decent manner. It does wrong to us to say that an exact alignment of the two figure remains impossible, as both operations were performed in different hospitals and were therefore also sent to different pathology departments, leading to the diversity in graphic presentation. Referring to comment #8 we tried to rearranged and enlarged the specimen images to provide better quality a most similar arrangement of the staining patterns.

  1. Use the past tense in all the text of case report.

Answer: Again, thank you for the attentive advice concerning grammatical tense rules - we adjusted the whole case report to past tense and marked the adapted verbs in red color.

  1. Rewrite the sentence on page 11 rows 406-409, because loss of expression of MMR and MSI do not confirm a LS.

Answer: A very important fact concerning the need of germline testing to diagnose LS in an affected patient - thank you very much for pointing out this incorrect statement. We removed the latter phrase and streamlined the sentence, also due to revisions concerning comment #8, which are marked in red in line 448 to 453.

Reviewer 2 Report

In this study, Lindner AK et al presented the Lynch syndrome-associated urothelial cancer of the upper urinary tract and bladder, their germline profiles and outcomes compared to sporadic urothelial cancer, the impact of genetic testing, as well as urological follow-up strategies in Lynch syndrome. In addition, they reported the case of metastatic Lynch syndrome-associated urothelial cancer of the upper urinary tract and bladder with an achieved complete response during checkpoint inhibition since more than 2 years. The manuscript is straightforward, well written, and concise, and has clear results within the scope of a review article. Definitely deserves to be published and is a valuable contribution to the “International Journal of Molecular Sciences”. Some minor flaws need to be addressed before publication.

Minor points:

[1]1.3. Lynch syndrome and cancer susceptibility”, Page 3/18, Lines 99-101:

In addition, the MSH6 pathogenic variant carriers seem to be a gender-specific factor in cancer susceptibility with higher EC risk, but only a low risk of colon cancer in both sexes [7].”.

At that point, please do clarify that mismatch repair (MMR) deficiency has been demonstrated in 20–40% of endometrial cancers, but data on its prognostic value are controversial.

Recommended reference: Boussios S, et al. Wise Management of Ovarian Cancer: On the Cutting Edge. J Pers Med. 2020;10(2):41.

[2]1.3. Lynch syndrome and cancer susceptibility:

At the end of this section, it would be beneficial for the readers to be incorporated into a table the cancer risk in Lynch syndrome by anatomical site (colon, endometrium, upper urinary tract, ovary, stomach, skin, bladder, etc).

[3]1.4.Diagnosis of Lynch syndrome”, Page 5/18, Lines 151-153:

In contrast to CRC where MSI is a well-known prognostic marker, the association between MSI and IHC in extra-colonic LS cancers becomes more and more weaker, suggesting that microsatellites are organ-specific [49].”.

Please, clarify that in the field of colorectal cancer, beyond the prognostic value of MSI status in patients with stage II disease, there is also predictive importance for the response to immunotherapy in metastatic setting.

Recommended reference: Boussios S, et al. The Developing Story of Predictive Biomarkers in Colorectal Cancer. J Pers Med. 2019;9(1). pii: E12).

[4]1.7. Lynch syndrome and immunotherapy”:

In this section, please, make a comment about the relevant status beyond colorectal cancer. Indeed, among epithelial ovarian cancers, high grade serous is the histologic subtype correlated with higher PD-L1 positivity and more tumor infiltrating lymphocytes.

Recommended reference: Demircan NC, et al. Current and future immunotherapy approaches in ovarian cancer. Ann Transl Med 2020. doi: 10.21037/atm-20- 4499.

[5] A workflow diagram for the study would be of benefit for the readers.

Author Response

Comments reviewer #2

  1. 3 Lynch syndrome and cancer susceptibility, Page 3/18, Lines 99-101: “In addition, the MSH6 pathogenic variant carriers seem to be a gender-specific factor in cancer susceptibility with higher EC risk, but only a low risk of colon cancer in both sexes [7].” At that point, please do clarify that mismatch repair (MMR) deficiency has been demonstrated in 20–40% of endometrial cancers, but data on its prognostic value are controversial. Recommended reference:Boussios S, et al. Wise Management of Ovarian Cancer: On the Cutting Edge. J Pers Med. 2020;10(2):41. 

Answer: Dear Reviewer, thank you for your precise comment and mentioning this detail to improve our review. As proposed, we included a sentence underlining the statement of MMR deficiency in EC and inserted the reference number 27. The whole sentence can be found in line 104 to 106 and is marked in red color.

  1. 3 Lynch syndrome and cancer susceptibility: At the end of this section, it would be beneficial for the readers to be incorporated into a table the cancer risk in Lynch syndrome by anatomical site (colon, endometrium, upper urinary tract, ovary, stomach, skin, bladder, etc).

Answer: A important and welcome suggestion to certainly improve our review, thank you very much. We implemented an overview figure (Figure 3) into paragraph 1.3, showing the individual cancer risk depending on anatomical site and referred to it in the text in line 120 to 122. Further, the numbering of the following figures in the manuscript were adapted and the new numeration marked in red.

  1. “1.4. Diagnosis of Lynch syndrome”, Page 5/18, Lines 151-153: “In contrast to CRC where MSI is a well-known prognostic marker, the association between MSI and IHC in extra-colonic LS cancers becomes more and more weaker, suggesting that microsatellites are organ-specific [49].” Please, clarify that in the field of colorectal cancer, beyond the prognostic value of MSI status in patients with stage II disease, there is also predictive importance for the response to immunotherapy in metastatic setting. Recommended reference: Boussios S, et al. The Developing Story of Predictive Biomarkers in Colorectal Cancer. J Pers Med. 2019;9(1). pii: E12).

Answer: Thank you for pointing out this important fact on MSI predicting the response to immunotherapy in CRC - this of course is of great relevance when it comes to the field of LS-associated tumors. We streamlined the sentence and adapted the phrase to implement the suggested statement as can be seen in line 184 to 186.

  1. “1.7. Lynch syndrome and immunotherapy”: In this section, please, make a comment about the relevant status beyond colorectal cancer. Indeed, among epithelial ovarian cancers, high grade serous is the histologic subtype correlated with higher PD-L1 positivity and more tumor infiltrating lymphocytes. Recommended reference: Demircan NC, et al. Current and future immunotherapy approaches in ovarian cancer. Ann Transl Med 2020. doi: 10.21037/atm-20- 4499.

Answer: Thank you for your constructive comment concerning PD-L1 expression and ovarian cancer, as this represents another important LS-associated tumor dignity. We are glad to include your suggested comment in our paragraph, highlighted in red from line 398 to 401 and thankfully added the reference to our literature, holding number 123.

  1. A workflow diagram for the study would be of benefit for the readers.

Answer: We thank the reviewer for this important quality factor for a good review. We included an overview workflow chart and included this in the paragraph ‘1.1 Materials and Methods’ as Figure 1, illustrating the number of search results and further assignment to the review literature.